# Dairy Cows Are Limited in Their Ability to Increase Glucose Availability for Immune Function during Disease

**DOI:** 10.3390/ani13061034

**Published:** 2023-03-12

**Authors:** Jonas Habel, Albert Sundrum

**Affiliations:** Department of Animal Nutrition and Animal Health, University of Kassel, Nordbahnhofstraße 1a, 37213 Witzenhausen, Germany

**Keywords:** energy balance, glucose balance, production diseases, immune defense

## Abstract

**Simple Summary:**

Both immune defense and milk synthesis require large amounts of energy, particularly from glucose, resulting in competition between these functions if glucose availability is limited. Based on comprehensive data on cow nutrient intake and on cow health records, the magnitudes and dynamics of glucose and energy balances of dairy cows kept on an experimental farm were evaluated in the weeks before, of, and after a production disease was diagnosed. Because dry matter intake (DMI) decreased during the phase of disease and was not adequately compensated by a reduction in milk yield (MY), glucose availability dropped when it was most needed.

**Abstract:**

Shortages of energy and glucose have been hypothesized to play a key role in the development of and responses to production diseases in dairy cows during early lactation. Given the importance of glucose for immune functions, we used a recently established method for the estimation of glucose balance (GB) to evaluate glucose availability during disease phases. A dataset comprising ration analyses as well as individual daily milk yields (MY), dry matter intake (DMI), body weights, and health records of 417 lactations (298 cows) was used to calculate individual daily GB and energy balance (EB). The magnitude and dynamics of MY, DMI, GB, and EB were evaluated in the weeks before, at, and after diagnoses of inflammatory diseases in different stages of early lactation from week in milk 1 to 15. Diagnoses were categorized as mastitis, claw and leg diseases, and other inflammatory diseases. Mixed linear models with a random intercept and slope term for each lactation were used to evaluate the effect of diagnosis on MY, DMI, GB, and EB while accounting for the background effects of week in milk, parity, season, and year. When unaffected by disease, in general, the GB of cows was close to zero in the first weeks of lactation and increased as lactation progressed. Weekly means of EB were negative throughout all lactation stages investigated. Disease decreased both the input of glucose precursors due to a reduced DMI as well as the output of glucose via milk due to a reduced MY. On average, the decrease in DMI was −1.5 (−1.9 to −1.1) kg and was proportionally higher than the decrease in MY, which averaged −1.0 (−1.4 to −0.6) kg. Mastitis reduced yield less than claw and leg disease or other diseases. On average, GB and EB were reduced by −3.8 (−5.6 to −2.1) mol C and −7.5 (−10.2 to −4.9) MJ in the week of diagnosis. This indicates the need to investigate strategies to increase the availability of glucogenic carbon for immune function during disease in dairy cows.

## 1. Introduction

Reducing the incidence of production diseases (PD) in dairy farming is of pivotal importance. They are of major importance for the economic viability of dairy farms [1,2]. They also raise public concerns about animal welfare [3] and undermine social acceptance of dairy farming. PD are multifactorial diseases. Their occurrence is the result of interactions between cow-specific factors, such as the cow’s endocrine and nutritional status, and numerous environmental factors such as climatic and hygiene conditions [4,5,6].

The metabolic load resulting from the increase in the milk synthetic capacity over the past decades has been put forward as a major risk factor for the occurrence of PD [7,8]. However, high milk yields do not necessarily imply an increased health risk, as nutritional imbalances and metabolic stress occur only when the performance is not met by an adequate energy and nutrient supply. Generally, the capacities for milk synthesis and feed intake are highly correlated [9]. However, due to increases in body weight and the corresponding maintenance requirements and due to limitations regarding the energy density as well as the time to eat and ruminate, increases in overall energy intake are not sufficient to meet the increased demand during early lactation of cows selected for high milk yields [6,10]. Moreover, selection for milk yield has led to higher feed conversion efficiencies for milk production, i.e., a higher amount of milk produced per unit of dry matter intake. Changes in this ratio are often evaluated as the “residual feed intake” (RFI), with low RFI being associated with higher feed efficiency for milk production [11,12]. Because feed costs are assumed to decline with low RFI [13], this criterion is often used for selection in dairy breeding [12,14]. However, metabolic imbalance and increased lipomobilization arising from strongly negative energy balance (NEB, a negative ratio between energy intake and energy requirement) is a major concern associated with low RFI because it increases the risk for the development of diseases [5,15,16,17]. Excessive lipomobilization causes impaired liver [18,19] and immune functions [20,21] following the hepatic accumulation of triglycerides as well as the increase in plasma levels of non-esterified fatty acids and β-hydroxy-butyrate. Besides metabolic imbalances, a low RFI also implies that the proportion of feed energy intake attributed to maintenance functions decreases. This effect is termed “dilution of maintenance” [22]. Thus, if cows are in NEB, a low RFI also implies that cows have fewer remaining energy sources and essential substrates available for functions other than milk synthesis.

Due to limitations in concentrate supply, rumen fill, and hepatic gluconeogenesis, competition for glucose in particular between the mammary gland and immune cells has been outlined [23,24]. Kvidera et al. (2017) investigated the drop in milk yield and the level of glucose infusion required to maintain euglycemia in mid-lactating dairy cows infused with lipopolysaccharide (LPS) and found that a fully activated immune system may need amounts of glucose similar to those required for the production of ~40 kg of milk [25].

While the amounts of glucose required for a specific inflammatory condition remain unknown under practical conditions, a methodology indicating the quantities of ‘residual’ glucose available to individual cows (which are expected to reflect a cow’s ability to respond to an inflammatory challenge) has been developed previously. The methodology is described in more detail elsewhere [26]. Briefly, it encompasses the estimation of the glucose demands of major glucose-consuming tissues (mammary gland, muscle, red blood cells, and the brain) as well as major sources of glucogenic supply (digestive and endogenous) from daily measurements of milk yield, body weight, and feed intake as well as the chemical composition of feedstuffs ingested. Here, we have applied the methodology to quantify the level as well as weekly changes (Δ) in the energy and glucose availability resulting from changes in DMI, MY, and body weight before, during, and after inflammatory disease events in dairy cows.

## 2. Materials and Methods

### 2.1. Animals

During and outside different experimental periods between January 2015 and February 2022, daily milk yields (MY, kg), feed intakes (kg), and body weights (BW, kg) of Holstein cows kept in freestall barns were recorded at the Educational and Research Centre for Animal Husbandry Hofgut Neumuehle. Experiments were part of the OptiKuh project [27], which includes feeding trials on phosphorus and nitrogen reduction, amino acid supplementation, and processing of corn. None of the trials intended to induce (NEB). Milk solids and body condition scores were assessed routinely (biweekly or monthly). The individual feed intake was recorded using feeders equipped with a weighing unit and automatic cow identification (Roughage Intake Control, Insentec B.V., Marknesse, The Netherlands). The cows were milked twice daily using a combination of a herringbone and a side-by-side milking parlor manufactured by GEA Farm Technologies (located in Bönen, Germany). The cows’ BWs were measured automatically after each milking via a walk-over scale, and daily values were derived by averaging morning and evening BW. Data from the milking parlor was recorded via the herd management system Dairy Plan C21 (GEA Farm Technologies, Boenen, Germany). The start of recording of feed intake, BW, and MY data of cows varied between days one and eight due to variation in the timing of entry after calving into the main housing system for lactation. The daily MY was recorded at morning and evening milking. The milk was analyzed for fat, protein, and lactose via an infrared analyzer (MilkoScan FT-6000, Foss Analytical A/S, Hillerod, Denmark; LKV Rheinland-Pfalz-Saar e.V., Bad Kreuznach, Germany).

Measurements of milk solids and BCS as well as values for MY and DMI that were missing due to technical errors were inter-/extrapolated linearly up to five consecutive days. The respective part of lactation was excluded from the analyses where more values were missing. After removal of values differing by more than 10 kg of empty body weight (EBW) from the previous or succeeding day (or up to 50 kg within 5 days), BW was smoothed across the first 150 days in milk (DIM) for each cow using a cow-specific polynomial function up to the fifth degree.

In total, data from 5048 weeks (Weeks 1 to 22 in milk corresponding to DIM 1 to 150) of 417 lactations of 298 cows were entered into the final dataset used in our analysis. Due to the variation in the timing of entry to the main herd after calving and due to technical constraints leading to missing records, weeks in milk 1 to 22 were not equally represented in the dataset. Of the 5048 weeks observed, only 98 (1.9%) and 214 (4.2%) were first and second weeks of lactation, respectively. Because some of the experimental recordings stopped at 105 DIM, the proportions of single weeks in milk between week 16 and 22 were also lower (between 2.5% and 4.5%) than those of weeks 3 to 15, which made up between 4.8% and 5.7% of the data. The curves for MY, DMI, GB, EB, and EBW followed the pattern typically observed in dairy cows during early lactation. To investigate differences in the way dairy cows adjust DMI and MY during disease, weeks 1 to 22 were divided into three stages of early lactation (stage 1: week 1 to 7, stage 2: week 8 to 14, and stage 3: week 15 to 22). Averages of MY, DMI, GB, EB, and EBW as well as their weekly change during different stages of early lactation are given in Table 1. 

### 2.2. Health Records

Diagnoses were made by a veterinarian according to a standardized diagnostic key that is used for the evaluation of health data in Germany [28]. Each diagnosis was classified in one of the following categories: udder disease, metabolic disease, claw and leg disease, genital tract disease, gastrointestinal disease, respiratory disease, and other diseases. Due to few diagnoses in categories other than udder disease and claw and leg disease, all other diagnoses were combined in the category “other diseases”. Given the hypothesis that substantial amounts of glucose are required for immunoactivation, diagnoses that were considered to cause no or minor/local inflammatory reactions, including dermatitis digitalis (DD) stadium M0, M1, M4, mild trichophytia, and ovarian cysts, were removed from the dataset. In the final dataset, all cases of descriptive mastitis diagnosis except those identified as sub-clinical and chronic built up the category “mastitis”. Apart from this, no further differentiation regarding the severity of mastitis cases was made. “Claw and leg disease” included ulcers, white line abscesses, interdigital phlegmons, acute DD stages, DD-associated inflammation as well as high-grade swelling of coronet and bulb, as they are supposed to be associated with the presence of infections [29]. In the category of “other diseases”, retained placenta, endometritis, metritis, vaginitis, and a few other infections, such as pneumonia, were compiled. In total, the number of inflammatory diagnoses made was 60, 34, and 26 for mastitis, claw and leg disease, and other diseases, respectively. This is equivalent to an incidence of 29 cases of inflammatory disease per 100 lactations or a 2.4% risk per cow-week (Table 2). Health records were cleaned stepwise by marking the day of the first diagnosis of each cow in each category as well as the following 10 days (“sick days”) with “1” and all other days with “0”. Second (third, fourth, and fifth) diagnoses of the same cow in the same category were considered only if the timing did not fall within the ”sick days” period of the preceding diagnosis to avoid double recordings of the same case of disease. In total, 83 cows were diagnosed once, 14 cows were diagnosed twice, and 3 cows were diagnosed three times within the first 22 weeks of lactation.

### 2.3. Rations

All cows were fed a total mixed ration (TMR) with varying compositions. All diets included grass silage, corn silage, and a mix of concentrates and were complemented by one or more of the following feedstuffs: pressed sugar beet pulp silage, hay, straw, vegetable oils, urea, and a mixture of synthetic amino acids (Table 3). Ration composition was adjusted regularly to ensure adequate levels of metabolizable energy intake. Cows were fed ad libitum. TMR and feedstuffs were analyzed monthly and/or if ration composition changed using Weende and van Soest analysis for dry matter, organic matter, crude nutrients (protein, fat, fiber, starch, and sugar) as well as ash-free acid detergent fiber and ash-free neutral detergent fiber. Means and standard deviations (SDs) of crude nutrient composition of the TMRs are given in Table 3.

### 2.4. Calculations

Daily energy demand for production (EDP; MJ NEL/d) and daily energy demand for maintenance (EDM; MJ NEL) were calculated according to the Society of Nutrition Physiology, which periodically publishes calculation principles for the evaluation of feedstuffs and rations typically used in the German dairy sector [30]. Daily energy intake was calculated from the energy content of the feed (MJ NEL) and the recorded individual dry matter intake (DMI; kg). The sum of EDP and EDM was subtracted from energy intake to obtain daily individual energy balance (EB; MJ NEL).

The main input variables for the calculation of GB were the individual daily milk yield, feed intake and body weight, and data on ration composition and chemical analyses of feedstuffs. Detailed explanations and equations for the calculation of glucose balance are reported elsewhere [26]. In brief, the calculations have a metabolic and a digestive part. In the metabolic part, the daily release of glucogenic C from endogenous sources is estimated from daily changes in empty body weight (ΔEBW, kg). EBW is calculated from the individual daily BW and the level of NDF intake (NDF%BW) according to INRA [31]. Additionally, the proportion of L-lactate production in protein tissue is taken into account according to a regression equation derived from data on the irreversible loss rate of glucose in protein tissue compiled by Larsen and Kristensen [32]. Glucose demands of major glucose-consuming tissues, including the mammary gland, protein tissue, the brain, and red blood cells, are calculated from the lactose yield (mammary gland) and individual body weight (protein tissue, brain, and red blood cells) according to previously published study in the field [33,34,35,36,37,38].

The estimation of the supply with glucogenic carbon (C) from feed is based on data-driven models that predict ruminal production of propionate as well as duodenal flows of L-lactate, glucogenic amino acids, and glucose from the contents of ruminal fermentable organic matter (RfOM), truly digestible starch (StDI), and truly digestible protein in the intestine (PDI) via the Systool Web application [39]. The calculations account for digestive interactions related to the proportion of concentrate in the diet, rumen protein balance, and the individual feeding level (feed intake in relation to body weight). In a second step, portal flows of glucose and glucogenic precursors were calculated from their ruminal and intestinal flows according to Loncke et al. [40] and Martineau et al. [41]. “Mol of glucogenic C per day” (mol C/d) was used as the general unit for the calculation of input and output fluxes as well as GB.

### 2.5. Statistical Analyses

The analyses were restricted to the first 22 weeks in milk (154 days) to focus on the early part of lactation when metabolic stress is most severe. Due to strong intra-individual variance from day to day, values were aggregated as weekly means. Two statistical approaches were used.

First, two-sided tests for difference of means between the mean value in a week (wk) relative to the week of diagnosis (wk-2, wk-1, wk0, and wk+1) and the mean of all other weeks (“healthy weeks”) were performed. The following variables were tested: GB, EB, MY, and DMI as well as the development of those variables compared to the preceding week (ΔGB, ΔEB, ΔMY, and ΔDMI). The test was performed for different disease categories as well as for different stages of early lactation (see above). Means in wk-2, wk-1, wk0, and wk1 differing significantly from healthy weeks are indicated at a level of *p* = 0.1 (*) and *p* = 0.05 (**). For the second approach, mixed linear models were used to estimate GB, EB, MY, and DMI as well as ΔGB, ΔEB, ΔMY, and DMI in wk-2, wk-1, wk0, and wk1 compared to healthy weeks. The modeling included repeated measures of lactation weeks within lactations following a first-order autoregressive moving average structure. Each of the models included a random intercept and slope term for each lactation with a covariance structure assuming no correlation between lactations. For the fixed effects of week in milk, polynomials of week in milk up to fourth degree representing the shape of the GB, EB, MY, DMI, and EBW curves were chosen by best fit according to the Akaike information criterion. To account for other confounding effects, the fixed effects of parity, season, and year were included. Finally, a derived variable of five levels representing the week relative to the week of diagnosis (“WeekDia”; two levels for the 2 weeks before the disease event, one level for the week of diagnosis, one level for the week after diagnosis, and one level for all other weeks) was built and added to the models. The model equation is: $$\begin{array}{c} y_{i,p}=\beta_{0}+\beta_{1}t_{i}+\beta_{2}{t^{2}}_{i}+\left(\beta_{3}{t^{3}}_{i}\right)+\left(\beta_{4}{t^{4}}_{i}\right)+{season}_{j}+{year}_{k}+{parity}_{l}+{WeekDia}_{m}+\alpha_{0}\\ \;\;\;\;\;\;+\alpha_{1,p}t_{i}+\alpha_{2,p}{t^{2}}_{i}+\varepsilon \end{array}$$
where $y_{i,p}$ is the ith weekly (t) GB (EB, MY, DMI, ΔGB, ΔEB, ΔMY, ΔDMI, and ΔEBW) measured on ${\mathrm{cow}}_{p}$; $\beta_{0}$ is a regression coefficient for the intercept; $\beta_{1}$, $\beta_{2}$, ($\beta_{3}$), and ($\beta_{4}$) are regression coefficients for the polynomial terms to estimate weekly values as a function of week in milk across all cows in the herd; ${season}_{j}$, ${year}_{k}$, ${parity}_{l}$, and ${WeekDia}_{m}$ represent the fixed effects of year, season, parity, and WeekDia on $y_{i,p}$, respectively; $\alpha_{0}$, $\alpha_{1}$, and $\alpha_{2}$ are the random intercept and slope terms to describe the deviation of ${\mathrm{cow}}_{p}$’s relative weekly values from that of the rest of the herd; and ε represents the random residual error. Residuals were checked graphically against predicted values to test the homogeneity of variance of the error terms. The mixed model was fitted using the mixed linear model procedure in IBM^®^ SPSS^®^ Version 28.0.1.0.

## 3. Results

On average, GB and EB increased from 3.9 ± 16.8 mol C/d and −73.5 ± 25.7 MJ NEL/d in the first week of lactation to 31.1 ± 17.8 mol C/d and −12.7 ± 21.3 MJ NEL/d in the fifteenth week of lactation.

When evaluated across all stages of early lactation, means of GB were lower in the week a disease was diagnosed (11.3 mol C/d; *p* =0.001) as well as in the week before diagnosis (13.7 mol C/d; *p* = 0.070) compared to the average of healthy weeks (16.7 mol C/d). Among diseased cows, GB was lowest in cows diagnosed with claw and leg disease. Means of MY were higher in the week before, of, and after diagnosis compared to other weeks during the same lactation stage, although the average daily MY declined by an average of 1.00 kg in the week of diagnosis compared to the preceding week (ΔMY). Means of DMI were higher in the week before and after diagnosis but lower in the week of diagnosis compared to healthy weeks during the same lactation stage. The average reduction in daily DMI in the week of diagnosis compared to the preceding week was 1.06 kg (ΔDMI).

When evaluated for different disease categories (Figure 1) as well as for different stages of early lactation (Figure 1 and Figure 2), however, differences in the way MY and DMI (and thus GB) adapted during disease events were observed.

In the first stage (weeks one to seven), average daily MY of diseased cows increased even in the week of diagnosis, although it increased more slowly than in healthy weeks during this stage (ΔMY averaging +0.3 vs. +1.3 kg; *p* = 0.061). The average daily DMI was reduced by −1.2 kg in the week of diagnosis compared to the preceding week (ΔDMI), while this value was +0.7 kg in healthy weeks during this stage (*p* = 0.000). This resulted in a more severe decrease in GB in the week of diagnosis compared to healthy weeks (−5.8 vs. −0.6 mol C; *p* = 0.022), with absolute values of GB averaging 0.9 and 7.4 mol C/d in the week of diagnosis and healthy weeks, respectively (*p* = 0.007). During the first stage, cows diagnosed with mastitis showed lower MY than cows diagnosed with claw and leg disease or other diseases. However, their average daily MY, DMI, and GB in the week of diagnosis did not change significantly compared to the preceding week and was not significantly different from the average MY, DMI, and GB observed in healthy weeks during this stage. Because their MY dropped in the week after diagnosis, GB in the week after diagnosis was greater than the average GB in healthy weeks during this stage. In contrast, GB was significantly lower in the week of diagnosis as well as in the week after diagnosis compared to healthy weeks in cows diagnosed with claw and leg disease or other diseases.

In the second stage investigated (weeks 8 to 14), MY decreased more significantly in the week of disease compared to healthy weeks, with ΔMY averaging −2.0 vs. −0.3 (*p* = 0.000). Because the decrease in DMI during disease was not as severe as in the first stage, with ΔDMI averaging −0.8 kg in the week of diagnosis compared to +0.3 kg in healthy weeks (*p* = 0.002), no significant differences in ΔGB, which, on average, became positive during this stage, were observed between the week of diagnosis and healthy weeks in this stage (1.6 vs. 1.9 mol C; *p* > 0.1). Accordingly, absolute values of GB in the week of diagnosis and healthy weeks averaged 13.9 and 16.9 mol C/d (*p* > 0.1), respectively. In the second stage, the drop in MY and DMI in the week of diagnosis was observed for all disease categories, but only cows diagnosed with claw and leg disease or other diseases had significantly lower absolute GB values when compared with healthy weeks during the same stage.

In the third stage (weeks 15 to 22), the decrease in MY in the week of diagnosis was not as strong as in the second stage, with ΔMY averaging −1.4 kg compared to −0.4 kg in the healthy weeks of this stage (*p* = 0.016). Because DMI decreased more significantly in the week of diagnosis than in the healthy weeks during this stage (ΔDMI averaging −1.2 vs. −0.1 kg; *p* = 0.006), GB stopped increasing in the week of diagnosis, with ΔGB averaging −1.9 mol C compared to +1.3 mol C in healthy weeks of this stage (*p* = 0.0.58). GB averaged 20.3 and 26.9 mol C/d in the week of diagnosis and heathy weeks, respectively (*p* = 0.018). GB of cows diagnosed with claw and leg disease in the third stage was significantly lower than the average of healthy weeks. This was due to both higher MY in the week before, of, and after diagnosis and similar (week before and after diagnosis) or lower (in the week of diagnosis) DMI. Although the MY of cows diagnosed with mastitis in the third stage was higher than the average MY in healthy weeks of this period, their GB was similar due to higher DMI.

Besides marginal differences in the absolute values and in the level of significance, EB followed a similar pattern to GB during disease. Pearson correlation coefficients for weekly means of GB, EB, DMI, MY, and ΔEBW across all weeks investigated are given in Table 4.

Due to the limited number of disease events in each category, the fixed effect of the week of diagnosis included in the mixed linear models (for GB, EB, MY, and DMI) included all diagnoses. Results of mixed linear modeling, which also accounted for the fixed effects of week in milk, year, season, and parity and included a random intercept and slope term for each lactation, showed that average daily GB, EB, MY, and DMI in the week of diagnosis was −3.8 (−5.6 to −2.1) mol C, −7.5 (−10.2 to −4.9) MJ, −1.0 (−1.4 to −0.6) kg, and −1.5 kg (−1.9 to −1.1), respectively, compared to weeks without diagnosis. When testing for the change in GB, EB, MY, and DMI compared to the preceding week while accounting for the same fixed effects, ΔGB, ΔEB, ΔMY, and ΔDMI in the week of diagnosis was −3.1 (−5.0 to −1.1) mol C, −4.7 (−8.0 to −1.5) MJ, −1.2 (−1.7 to −0.8) kg, and −1.4 (−1.9 to −0.9) kg, respectively, when compared to weeks without diagnosis. Although ΔGB, ΔEB, ΔMY, and ΔDMI became positive in the week after diagnosis, overall GB and EB did not recover as coefficients were still negative at −1.7 (−3.3 to 0.0) mol C/d and −3.4 (−5.9 to −0.9) MJ/d, respectively, compared to weeks without diagnosis. Results of the models for GB, EB, MY, and DMI as well as for ΔGB, ΔEB, ΔMY, and ΔDMI are given in Table A1 and Table A2.

## 4. Discussion

Due to a reduced DMI and the associated limited availability of nutrients in the digestive tract and in the intermediary metabolism during early lactation, metabolic trade-offs exist between productive and other life functions, such as reproductive and immune functions, particularly in high-yielding dairy cows [23,42]. Severe negative energy balance resulting from the mismatch between food energy intake and energy expenditure [5,43] and the severe loss of body tissue mass, i.e., the change in EBW or the change in body condition scores, which is a result of this mismatch, have been brought forward as risk factors for the development of PD in dairy cows [7,44]. It has also been shown that metabolic adaptations to similar levels of NEB differ greatly between individual cows [45]. Because glucose is the central metabolite for both mammary and immune cells, competition for this specific metabolite is at the core of the metabolic conflict [24]. Nevertheless, this conflict has not been addressed intensively in dairy research so far. In this study, a methodology for the quantification of residual amounts of glucose, which includes both the amount of glucose derived from nutrient intake as well as the amount of glucose derived from the intermediary metabolism (from the change in EBW and from the main pathway of glucose recycling via lactate), was applied to evaluate the development of glucose balance of dairy cows during disease.

Although the metabolic burden imposed by the onset of lactation is supposed to be the starting point for subclinical and clinical metabolic disorders and, subsequently, other diseases, it is not predictable, if, at what time, and how animals respond to metabolic stress [6]. Due to the multifactorial character of PD, scientific evaluation of the relationship between individual nutrient availability and the occurrence of PD is difficult. Even if cow-individual data for DMI, MY, BW, and health status are collected in a consistent manner while housing and living conditions are highly standardized, this does not prevent large inter- and intra-individual variations in nutrient supply and other factors, such as the level of exposure to biotic and abiotic noxes or social stress, and the individual capacities and coping strategies [46]. Genomic and metabolomic research investigating, e.g., individual differences in tissue-specific mRNA expression and milk biomarkers, is thought to advance understanding of why animals respond so differently to similar stresses or are able to regenerate differently under identical conditions. [47,48]. However, knowledge of the level of individual reserves is of central importance, as these levels are required for the verification of any (genomic, nutritional, or management) effect. Thus, the cow-specific variation in glucose availability during and outside periods of disease may be linked more directly to individual differences in adaptability.

The incidence of disease recorded in our dataset is lower than what has recently been observed in a large sample of German dairy farms [49]. Generally, comparison of incidences is difficult due to great variation between farms, herds, and the methods and definitions used for disease recording. However, cows enrolled in this study were kept on an experimental farm, and the rather low incidence is likely to be due to the selection criteria applied (inflammatory disease only and exclusion of diagnosis within 10 days after the previous diagnosis) and the proportionally lower number of first and second weeks in milk recorded in our dataset. 

In our study, the occurrence of disease was associated with reductions in both milk yield and dry matter intake. Hypophagia during inflammation is a well-known phenomenon and is observed across many species [50,51]. In various studies, dairy cows diagnosed with mastitis, metabolic, or other diseases showed a reduced DMI compared with healthy cows [17,52,53,54]. Host cytokines such as tumor necrosis factor-α and interleukin−1β, and bacterial endotoxins [55,56] have been found to exhibit appetite-depressing effects.

While dry matter intake decreased significantly in the week of diagnosis throughout all lactation stages investigated, reductions in MY were low during early lactation despite an insufficient nutrient supply in relation to the requirements. In contrast, the ability of dairy cows to reduce milk production during disease was greater in later compared to early lactation stages. This is in accordance with a greater ability to reduce milk yield during nutritional challenges in mid vs. early lactation observed in other studies (e.g., [57]). It has been shown repeatedly that milk yield decreases during disease [58,59]. With regard to the rather low reductions in MY and DMI in cows diagnosed with mastitis observed in our study, interpretation is difficult due to the lack of differentiation of mastitis diagnoses, which is a weakness of this study. However, it can be assumed that the majority of diagnoses were mild mastitis cases and that they were identified and treated at an early stage, with inflammation being limited locally and being of short duration (i.e., drop in milk yield and recovery within a few days instead of weeks). In fact, the ability to reduce milk yield during mastitis or other diseases is not only affected by the stage of lactation but also by the level of inflammation and by the genetic merit for milk production. Endocrine changes such as peripheral insulin resistance and downregulation of hepatic growth-hormone receptors [60,61] favoring the flow of glucose to the mammary gland during the periparturient period [62] are physiologic but are more severe in cows bred for high milk yields [61]. According to our results, it has been shown that reductions in milk synthesis during disease in early lactation are rather low [57,63] even when challenged by intramammary inflammation [64,65].

In all lactation stages investigated, decreases in milk yield did not lead to increased GB and EB, i.e., to greater nutrient availability for self-sustaining life functions. Milk yield reductions were not sufficient to cover the reductions in energy and glucose supply emerging from decreased DMI. Results obtained from the mixed models indicate that the average daily GB was −1.1 to −5.0 mol C (95% CI) in the week of diagnosed diseases. This means that the glucose availability for self-sustaining life functions decreased in periods when glucose was urgently needed. To avoid this decrease in GB following the imbalance between milk yield reductions and reductions in dry matter intake in the week of diagnosis, an infusion of 34 to 151 g of glucose (equivalent to ~700–3000 mL of a 5% glucose solution or ~85–375 mL of a 40% glucose solution), a supplementation of 28–124 g of propionate, or an additional reduction of ~0.5 to 2.1 kg milk yield would have been required on average. Besides therapeutic options to increase the supply with glucose or glucogenic precursors in case of disease, increasing overall glucose availability, and, in particular, glucose availability for functions other than milk synthesis through nutritional interventions, is limited. Overall energy intake is restricted because of the risk of rumen acidosis in case of excessive intake of highly digestible carbohydrates, time to eat, rumen volume, and liver function. Moreover, increases in DMI or in the energy density of the diet during early lactation results in increases in milk energy output at a similar magnitude, with no beneficial effects on energy reserves for functions other than milk synthesis [66]. Regarding the nutrient composition of diets, it has been suggested that feeding glucogenic instead of lipogenic sources of energy favors the allocation of energy towards functions other than milk synthesis, although results are inconsistent [66,67,68,69]. However, avoiding overfeeding in the dry period was shown to alleviate metabolic imbalance related to the carbohydrate metabolism, such as insulin resistance, during early lactation [19].

With regard to the limitations of increasing energy intake and the supply with glucogenic C, and with regard to the inability of cows to sufficiently reduce milk yield during disease, we emphasize that it is possible to reduce milk withdrawal through incomplete milking during phases of disease and severe undernutrition. By supporting the physiological processes of nutrient reallocation in case of disease in this way, the economic loss in revenue from milk sales appears to be of minor importance if, at the same time, the costs of a severe case of disease are avoided. With regard to the risk of mastitis, it can be assumed that an amount of residual milk between 200 and 800 mL per quarter is not related to the incidence of mastitis [70,71,72,73], whereas inconsistent effects of a prolonged milking interval on the incidence of mastitis have been described [74,75,76].

The low reductions in milk yield despite an inflammatory disease, particularly during early lactation, indicate that dairy cows have limited ability to repartition glucose away from the mammary gland. To avoid negative effects on the health following the failure to simultaneously supply productive and self-sustaining life functions, an animal’s ability to metabolically adapt to additional demands (a trait which has been termed ‘plasticity’ [77]) is of major importance. It has been hypothesized that cows with high genetic merit for milk production have a reduced capacity to adapt partitioning of energy and essential substrates in response to additional demands [77]. Although this may explain the low reductions in milk yield and overall reduced availabilities of glucose and energy during disease observed in our study, the severity of disease, the degree of immunoactivation, and thus the demand from immune cells of diseased cows enrolled in this study are not known. Moreover, inflammation induces several metabolic adaptations other than milk synthesis reductions that increase glucose availability during disease which cannot be evaluated by means of GB calculation. Among them, increased glucose removal from the plasma pool [25,78], a shift in glucose transporter expression [79], and the depletion of glycogen stores [43] may increase glucose availability to immune cells to some degree. A reduced energy demand from the digestive tract following hypophagia, as reflected by reduced cell migration and turnover [80,81], may also increase glucose availability for other tissues. In total, however, the contribution of glucogenic C by these adaptations is expected to be low in dairy cows during early lactation, as glycogen depots are generally exhausted after calving [82,83], while homeostasis of plasma glucose is tightly regulated [84]. Hence, the low absolute residual amounts of glucose (on average, less than 20 mol C, which is equivalent to ~600 g of glucose) observed in our study indicate that dairy cows often do not have sufficient glucose available to respond to infections where they do not reduce milk synthesis adequately. Together with low metabolic plasticity, this points to the risks associated with the trend of increasing feed conversion efficiency for milk production. Besides the risk associated with increased levels of metabolic stress following severe NEB [85], it can be assumed that an insufficient supply of self-sustaining life functions with energy and glucose is a major threat to the health and, ultimately, the longevity of high-producing dairy cows [6]. With regard to failure costs of disease events, such as reduced milk yield, discarded milk, medication, labor, and in particular, premature culling [1], monetary gains from high feed efficiency for milk production may thus be offset by monetary losses when the cows’ ability to fuel immune functions is compromised. A sound economic evaluation of biological efficiencies should thus include all costs and benefits attributable to the service life of individual cows and herds, including not only feed costs but also the costs of disease and involuntary culling [2].

## 5. Conclusions

During early lactation, high-yielding dairy cows generally face a shortage in glucose for functions other than milk synthesis, such as the immune system. During phases of diseases, shortage in glucose increases as dry matter intake decreases more than milk yield. Further research on overall glucose balance and the effect of management measures such as the reduction of milk withdrawal, infusion of glucose, or supplementary feeding of propionate to cows challenged by inflammatory diseases should be envisaged.

## Figures and Tables

**Figure 1 animals-13-01034-f001:**
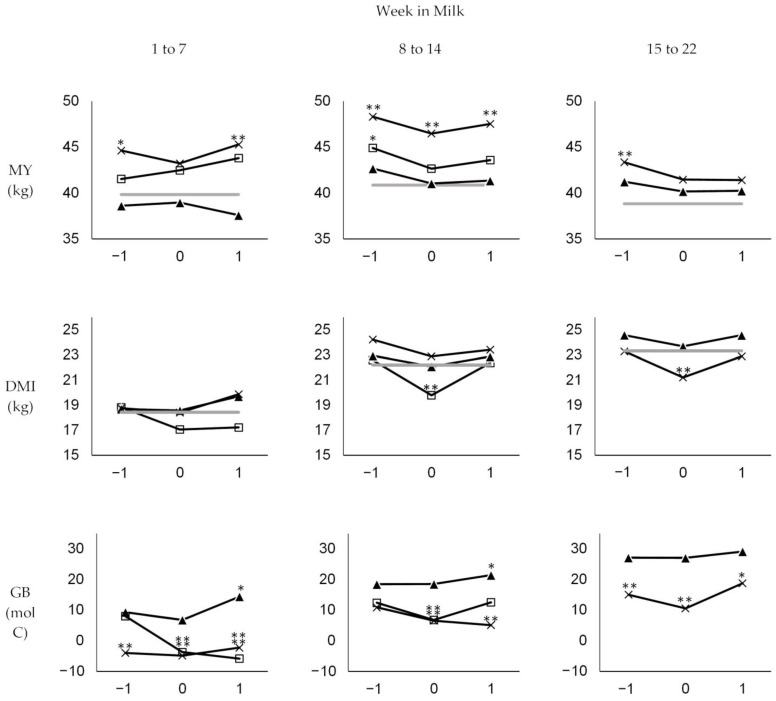
Weekly means of daily milk yield (MY), dry matter intake (DMI), and glucose balance (GB) in the week before (−1), the week of (0), and the week after (1) diagnosis in different stages of early lactation (weeks 1 to 7, 8 to 14, and 15 to 22) for different disease categories: claw and leg disease (x), mastitis (▲), and other diseases (□). Significant differences between means in the respective week compared to the means in healthy weeks of the respective stage (grey line) are indicated at *p* < 0.1 (*) and *p* < 0.05 (**).

**Figure 2 animals-13-01034-f002:**
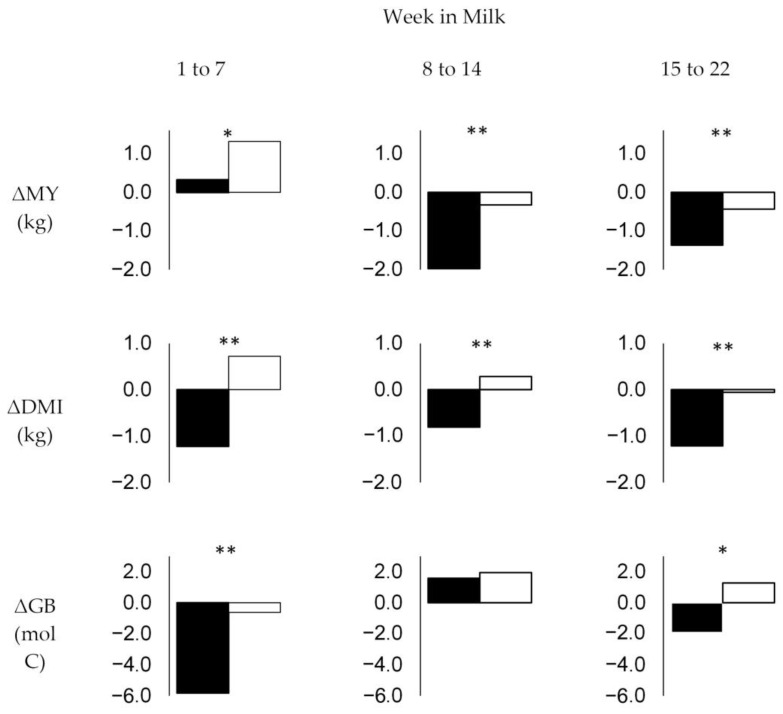
Weekly change (Δ) in means of daily milk yield (MY), dry matter intake (DMI), and glucose balance (GB) in the week of diagnosis (black bars) compared to the means of MY, DMI, and GB in other weeks of the respective stage (white bars) for all disease categories in different stages of early lactation (weeks 1 to 7, 8 to 14, and 15 to 22). Significant differences are indicated at *p* < 0.1 (*) and *p* < 0.05 (**).

**Table 1 animals-13-01034-t001:** Means of milk yield (MY, kg/d), dry matter intake (DMI, kg/d), glucose balance (GB, mol C/d), energy balance (EB; MJ of net energy for lactation (NEL)), and empty body weight (EBW, kg) as well as means of the weekly change (Δ) in average daily MY, DMI, GB, EB, and EBW during different stages of early lactation (data of all cows and lactations enrolled in the study).

Week in Milk			
	1 to 7	8 to 14	15 to 22
MY (kg)	40.0 ± 0.2	41.1 ± 0.2	39.0 ± 0.2
ΔMY	1.3 ± 0.1	−0.3 ± 0.0	−0.4 ± 0.1
DMI (kg)	18.5 ± 0.0	22.2 ± 0	23.4 ± 0.0
ΔDMI	0.7 ± 0.1	0.3 ± 0.1	0.0 ± 0.1
GB (mol C)	7.2 ± 0.4	16.7 ± 0.4	26.6 ± 0.4
ΔGB	−0.8 ± 0.3	1.9 ± 0.2	1.3 ± 0.2
EB (MJ NEL)	−53.6 ± 0.6	−31.2 ± 0.5	−17.4 ± 0.5
ΔEB	0.8 ± 0.6	3.0 ± 0.3	1.3 ± 0.4
EBW (kg)	564 ± 2	567 ± 1	585 ± 2
ΔEBW	−0.7 ± 0.0	0.1 ± 0.0	0.1 ± 0.0

**Table 2 animals-13-01034-t002:** Number of diagnoses according to disease category and stage of lactation.

	Week in Milk		
	1 to 7	8 to 14	15 to 22
Mastitis	17	25	18
Claw and Leg Disease	11	10	13
Other Disease	13	12	1

**Table 3 animals-13-01034-t003:** Mean, minimal, and maximal contents of dry matter (DM), net energy for lactation (NEL), organic matter (OM), crude protein (CP), ash-free acid detergent fiber (ADFom), and ash-free neutral detergent fiber (aNDFom) in the rations fed (A) as well as mean, minimal, and maximal proportions of feedstuffs in the diets (B).

A			
Variable	Mean	Min	Max
DM (g/kg fresh matter)	468	383	566
NEL (MJ/kg DM)	7.0	6.5	7.3
OM (g/kg DM)	929	918	939
CP (g/kgDM)	152	121	165
Starch (g/kgDM)	197	129	235
aNDFom (g/kgDM)	341	252	375
ADFom (g/kgDM)	213	148	232
**B**			
**Feedstuffs**	**Mean**	**Min**	**Max**
Concentrates	35.1	29.6	40.1
Maize silage	25.5	15.0	45.5
Grass silage	23.0	9.7	34.3
Sugar beet pulp silage	13.8	0.0	19.0
Hay	4.6	3.1	7.5
Barley straw	1.6	0.0	2.4
Urea	0.4	0.3	0.4
Amino acids	0.4	0.4	0.5
Vegetable oils	0.8	0.8	0.9

**Table 4 animals-13-01034-t004:** Pearson correlation coefficients (upper side) and their *p*-values (lower side) for weekly means of glucose balance (GB), energy balance (EB), dry matter intake (DMI), milk yield (MY), and the weekly change in empty body weight (ΔEBW) across all lactation weeks investigated.

	GB	EB	MY	DMI	ΔEBW
GB	1	0.861	−0.366	0.496	0.127
EB	0.000	1	−0.441	0.520	0.322
MY	0.000	0.000	1	0.526	−0.054
DMI	0.000	0.000	0.000	1	0.247
ΔEBW	0.000	0.000	0.000	0.000	1

## Data Availability

Restrictions apply to the availability of these data. Data were obtained from the Educational and Research Centre for Animal Husbandry “Hofgut Neumuehle” and are available with the permission of third party.

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
