# Peer review of "Dairy Cows Are Limited in Their Ability to Increase Glucose Availability for Immune Function during Disease"

_animals, 2023, doi:10.3390/ani13061034_

Round 1

Reviewer 1 Report

abstract

L31

 Mastitis reduced yield less than other diseases…. Include.  Which ones

 Disease also decreased the input of glucose precursors due to a reduced DMI. On average, the decrease  in DMI was proportionally higher than the decrease in MY reducing GB and EB during the phase 33 of disease when glucose requirements are assumed to increase.   Give amount ( data ) of  decrease DMI  vs  MY. Etc …

Table 1  corrects

netto energy for lactation (NEL),

TM ?

  ADF Goes after NDF

In Feedstuffs  correct. Values and gibe them in g/kg.  to formulate. 1 ton as DM,  specify. In footnote Amino acids  and Vegetable oils

L 242- 247

Verify that the text correspond. With the  tables results…. And. Explain better this paragraph

First week of lactation and increased up to 31.1 ± 17.8 mol C/d and -12.7 ± 21.3 MJ NEL/d 243 at the end of the 150-day period. 244

Means of GB were lower in the week a disease was diagnosed (11.3 mol C/d; p =.001) 245 as well as in the two weeks before diagnosis (12.5 and 13.7 mol C/d; p=.022 and .070) com-246 pared to the average of other weeks (16.7 mol C/d), when evaluated for all diseases (Figure 247 1).

Correct Table. 3

Add. On the upper  side. P values and  in the lower  side. The.  correlation ship  the value 1. Take it if,  it. is easer to  understand. P value. Vs. correlation

In general, the discussion needs to be improved

Reviewer 2 Report

The manuscript entitled “Availability of glucose for immune function is reduced by disease in dairy cows” deals with a very interesting problem of modern dairy production and merits serious consideration. Unfortunately, a major revision is required, mainly because the results of the study are inadequately presented. Issues regarding the Materials and Methods section are minor but they should be addressed.

Please consider the following comments:

TITLE

As written, I believe it focuses on the reduced dry matter intake, induced by disease. Considering that disease itself increases the need for enhanced immune function, as well, this title appears kind of “limiting”. Would the authors like to reconsider the title, including all aspects of this very important issue?   

SIMPLE SUMMARY

Fine

ABSTRACT

L22: Add number of cows.

L30-34: Some form of the actual results must be included.

L35: “to reduce the loss of glucose in milk”, this is a controversial issue, please see comments in the Discussion section

KEYWARDS

The right ones were chosen.

INTRODUCTON

L40: Reducing “the incidence of” production diseases is probably better.

L51-52: This sentence may not be technically accurate and should probably be rephrased; in both NRC 2001 and NASEM 2021, milk yield (whatever the unit may be) is used to estimate dry matter intake in a very straight forward way. Unit of milk times 0.372 or 0.305, respectively; therefore, if milk yield increases 30%, the part of dry matter intake attributed to milk production increases 30% also. It is the part of DMI attributed to BW (or LW) that does not change. Perhaps, using the word “overall” before feed intake and deleting “in step” will do the job, in any case, the authors should choose the best way to express their thoughts.

L65-67: The «dilution effect» does not imply “that cows have fewer remaining energy sources and essential substrates available for functions other than milk synthesis”. Consider two cows: Cow A, producing 40 L of milk per day and cow B producing 20 L of milk, both at a perfect EB. The “dilution effect” is there for cow A but being at EB her maintenance requirements are covered, as well. Please, rephrase.

L73-76: This sentence is difficult to follow; the part “led us to consider….” seems awkward considering the authors’ previous work (reference no 24). Please, rephrase.

L77-79: What does “the methodology used here” stands for in this sentence? Totally misplaced, consider rephrasing.

L82: The methodology to estimate/calculate the quantities of “residual” glucose available to individual cows, which encompasses……the rest is fine.

MATERIALS AND METHODS

L92: Does “During different experimental periods” mean that cows were used for other experiments and data collected were used retrospectively for the present objective (perfectly OK if negative energy balance was not induced) or that this was the experiment conducted in the first place, during different time periods? Was the OptiKuh project included? Some more details are needed here.

L94: “loose pen” is not proper use of English language.

L126: “diagnostic key”? More details are needed here (especially for mastitis).

L128: Considering diseases described in L139, I would use “genital tract” instead of “fertility” here.

L157(Table 1): “netto” or net?

L163: Non-German readers may not recognize GfE; please give some explanations.

L173: (NDF%BW; kg DM/kg BW) difficult to follow.

L193: Week 22 ends on day 154.

L197-200: Authors should probably improve syntax here.

L202: “and p=0.05 (**). or the second approach” Obviously, something is wrong here.

L211-214: Is the variable presented here, the “WeekDia” appearing later in the model? If yes, add the abbreviation to the text, please.

RESULTS

L230-244: From a technical point of view, it is the opinion of this reviewer, that these are not actually the results of the present study, as they are not related to the specific objectives sited in the Introduction section. You are not trying to find what the number of cases or what the incidence was. Nor which the production parameters were. I consider them a description of your data base and I believe that they should be moved to the appropriate Materials and Methods lines. Moreover, a) how many cows, out of the total of 298 had these 120 cases? And b) data in Table 2 apply for the whole 298 cows, not the sick ones; please make it clear in the text. By the way, stages (weeks 1-7, 8-15 and 15-22 are presented here for the first time. Clearly, this is Materials and Methods. Nowhere in the relevant section. Why is this classification used? (This doesn’t mean I disagree) And how many cases in the respective stages? Any thoughts regarding including this in the statistical model?

L245-248: The 11.3 mol C/d represents all diseases on week of diagnosis, if I understand it correctly. And 16.7 mol C/d represents “healthy” weeks. Don’t all values in Figure 1 seem to be lower than 10.0 mol C/d? OK, my mistake, this is for weeks 1-7, only. But then, for the other two stages, values seem quite high, can the mean be only 11.3 mol C/d? Briefly, the presentation of your results is, unfortunately, not up to the level of your work and to the interest it will generate to the readers. Few data and few comments, difficult to follow in L248-254, as well. Please see following comment.

L262-277: Presentation of results below expectations here, also. I am afraid, the Results section needs a major revision. It is all too complicated with weeks and stages mixed up. I would suggest to consider each stage separately. Don’t be afraid to make this section longer, no editor would object to a clear presentation of so interesting results based on word limitations (I guess…). Figures are fine, a much more supportive test is needed.

L284-290: Text adequate but without any idea regarding the severity of mastitis cases, results difficult to interpret. See also previous comment on diagnosis, regarding mastitis.

L295-309: Description of results OK. Are you sure that “increased” in L306 accurately describes what you want to state? I believe that some readers may misinterpret it. 

DISCUSSION

L314-318: Using “both” at the beginning of the sentence implies that these are two different phenomena while they are the same one. I suggest rephrasing.

L318: I don’t see the point for “nonetheless” here.

L321: Why “supposed to be”? Are there any opposite opinions? And delete the extra “space” after “competition”.

L337-340: Would genomics be of any help here?

L342-347: And disease definitions and disease recording methods, as well!

L352-354: This highlights the need to describe diseases in detail (especially mastitis).

L354-357: This is a very controversial issue. For the time being, as the authors of reference (50) report, there is no such evidence in ruminants. We have all heard about iron and Gram- bacteria, lactose and protein in diarrheic calves etc; most are speculations to say the least.  I suggest to delete this part.

L372: “periods”. This has drawn my attention throughout the text, as it is often used interchangeably with “stage”. Please, examine the text carefully and select one way to address the “1-7, 8-14 and 15-22 week” issue. It is so confusing for the reader. If you need the word “period” to describe something else, it is OK, but stick to “stage” for the specific issue.

L382-388: So, you suggest to not fully milk sick cows in order to overcome the negative glucose balance. I assume, that by leaving 0.4-0.5 L per quarter, milk synthesis will stop earlier since the pressure inside the udder will reach the mark of the “1/3 of the arterial pressure” earlier, as well. Quite interesting! Unfortunately, I was not able to find the reference (60) you use to support this practice, supposedly being harmless for cows, in regard to mastitis incidence (even if I had found it, I can use the German language for every-day tasks but not for scientific work). I am very reluctant to accept this, but I am willing to discuss it; still, I am not sure whether a relative reference is available in the English literature but I will gladly accept that I am wrong. Please elaborate on this issue.

L394: The parenthesis must “close” at some point.

L398: What do “availabilities” refer to?

Overall, the Discussion section is well-written but I suspect that if the Results section is more thoroughly presented, the authors may want to add some new material to it. This is by no means “compulsory”, though. I would like to have the authors’ opinion regarding the nutritional aspects of the GB issue. Are there components of the diet that will enhance/improve this balance? Do we know anything about this? I mean the overall herd diet (or the “fresh cow” diet), as well as the diet of animals recovering from disease (clinical nutrition). I believe this merits a (short) paragraph in the Discussion section.  

CONCLUSIONS

L429-432: This sentence has no place in the Conclusions section. It is an unproven suggestion (a speculation), it does not result from this study. This is something for future investigation as is still the whole issue of GB.

Round 2

Reviewer 2 Report

Manuscript greatly improved and all issues adequately addressed. The discussion section now clearly describes your findings and despite the "many numbers", results are there for whoever wants to study them. Still not persuaded from your references about the milk removal issue (old data, low milk production, low numbers, different objectives) but at least now it is presented in a less absolute way.